# Beyond Nutrient Deficiency—Opportunities to Improve Nutritional Status and Promote Health Modernizing DRIs and Supplementation Recommendations

**DOI:** 10.3390/nu13061844

**Published:** 2021-05-28

**Authors:** Michael I. McBurney, Jeffrey B. Blumberg, Rebecca B. Costello, Manfred Eggersdorfer, John W. Erdman, William S. Harris, Elizabeth J. Johnson, Susan Hazels Mitmesser, Robert C. Post, Deshanie Rai, Leon J. Schurgers

**Affiliations:** 1Department of Human Health and Nutritional Sciences, University of Guelph, Guelph, ON N1H 0B5, Canada; 2Friedman School of Nutrition Science and Policy, Tufts University, Boston, MA 02111, USA; jeffrey.blumberg@tufts.edu (J.B.B.); elizabeth.johnson@tufts.edu (E.J.J.); 3Center for Magnesium Education and Research, Pahao, HI 96778, USA; rbcostello@earthlink.net; 4Department of Internal Medicine, University Medical Center Groningen, 9713 GZ Groningen, The Netherlands; dr.eggersdorfer@gmail.com; 5Department of Food Science and Human Nutrition, University of Illinois, Urbana-Champaign, IL 61801, USA; jwerdman@illinois.edu; 6Department of Internal Medicine, University of South Dakota, Sioux Falls, SD 57105, USA; wsh@faresinst.com; 7The Fatty Acid Research Institute, Sioux Falls, SD 57106, USA; 8Science & Technology, Pharmavite LLC, West Hills, CA 91304, USA; smitmesser@pharmavite.com; 9FoodTrition Solutions, LLC, Hackettstown, NJ 07840, USA; robert.post@foodtritionsolutions.com; 10Global Regulatory and Scientific Affairs, Omniactive Health Technologies, Morristown, NJ 07960, USA; d.rai@omniactives.com; 11Department of Biochemistry, Cardiovascular Research Institute Maastricht (CARIM), University of Maastricht, 6200 MD Maastricht, The Netherlands; l.schurgers@maastrichtuniversity.nl

**Keywords:** DRIs, lutein, EPA and DHA, magnesium, vitamin K, dietary guidelines, nutritional status

## Abstract

The US Dietary Guidelines for Americans (DGA) provide dietary recommendations to meet nutrient needs, promote health, and prevent disease. Despite 40 years of DGA, the prevalence of under-consumed nutrients continues in the US and globally, although dietary supplement use can help to fill shortfalls. Nutrient recommendations are based on Dietary Reference Intakes (DRIs) to meet the nutrient requirements for nearly all (97 to 98 percent) healthy individuals in a particular life stage and gender group and many need to be updated using current evidence. There is an opportunity to modernize vitamin and mineral intake recommendations based on biomarker or surrogate endpoint levels needed to ‘prevent deficiency’ with DRIs based on ranges of biomarker or surrogate endpoints levels that support normal cell/organ/tissue function in healthy individuals, and to establish DRIs for bioactive compounds. We recommend vitamin K and Mg DRIs be updated and DRIs be established for lutein and eicosapentaenoic and docosahexaenoic acid (EPA + DHA). With increasing interest in personalized (or precision) nutrition, we propose greater research investment in validating biomarkers and metabolic health measures and the development and use of inexpensive diagnostic devices. Data generated from such approaches will help elucidate optimal nutrient status, provide objective evaluations of an individual’s nutritional status, and serve to provide personalized nutrition guidance.

## 1. Introduction

The second Sustainable Development Goal of the World Health Organization recognizes that nutrition is the foundation of peaceful, secure, and stable societies and the need for better nutrition to improve health and end poverty. While the association of dietary patterns with health is generally accepted, the complexity of the relationship led to the US developing the *Dietary Guidelines for Americans* (DGA) in 1980. Based on the facts that: (1) “about 40 different nutrients to stay healthy, these include vitamins and minerals, as well as amino acids (from proteins), essential fatty acids (from vegetable oils and animal fats), and sources of energy (calories from carbohydrates, proteins, and fats)” and (2) “These nutrients are in the foods you normally eat” [1], the DGA identified seven principles of a healthful diet with the goal of helping reduce nutritional deficiencies and risk of related illnesses. As mandated by Congress in 1990, new DGAs have been issued every 5 years with the 9th edition being released in 2020 [2]. This paper reviews the history of dietary guidance with respect to dietary supplements, nutritional contributions from food and dietary supplement use, and identifies opportunities to update the Dietary Reference Intakes (DRIs) for magnesium and vitamin K and establish DRIs for lutein and the omega-3 fatty acids, eicosapentaenoic acid (EPA) and docosahexaenoic acid (DHA).

## 2. Dietary Guidelines History

After the first DGA in 1980, a Dietary Guidelines Advisory Committee (DGAC) has convened every 5 years to assess the best available nutrition and health science and produce a scientific report used in the development of the DGA. The DGA are the cornerstone of US nutrition policies and programs, including food assistance and consumer education programs. The purpose of the DGA has evolved over the last 40 years and it now serves as a reference for regional, state, and local organizations, provides information to health professionals and healthcare systems, and functions as a call to action for food product innovators. The role of dietary supplements in national dietary guidance has been given only modest attention in each DGA edition. 

The DGA process has changed from guidelines decided by a group of experts to a synthesis of evidence-based reviews, data modeling and analyses to inform conclusions and implications resulting in a comprehensive scientific report of recommendations. In turn, this external Report forms the basis for Federal policy—the DGA—which gives advice for healthy eating patterns applied by the programs of all agencies with food and public health missions. The 1980 DGA recommended that women in their childbearing years may need iron supplements and women who are pregnant or breastfeeding may need more iron, folic acid, vitamin A, and calcium but “rarely need to take vitamin or mineral supplements if you eat a wide variety of foods” [1]. The 1985 DGA cautioned against consuming excessive amounts of any nutrient and state “large dose supplements of any nutrient should be avoided” with key exceptions, e.g., iron supplements of women of childbearing years and acknowledgement for “the need for more of many nutrients” during pregnancy and breastfeeding [3]. Key legislative actions in the 1990s influenced subsequent DGAs, many of which had impacts on nutrient intake needs and recommendations. For example, in 1992, the Department of Health and Human Services (HHS), the Public Health Service, and Centers for Disease Control (CDC) recommended all women of childbearing age consume 400 µg of folic acid daily through fortification, supplementation, and diet to prevent neural tube defects [4]. The Nutrition Labeling and Education Act (NLEA) (Public Law 101-535) and recent regulatory amendments [5,6] mandated nutrition labeling requirements on most foods sold at retail [7] including essential nutrients. The US Food and Drug Administration (FDA) proposed a health claim for folic acid and neural tube defects in 1994 [8] and mandated folic acid fortification of cereal grains in 1996 [9] that was subsequently expanded to include corn masa flour [10]. The Dietary Supplement Health and Education Act of 1994 (DSHEA) defined dietary supplements, labeling and manufacturing practices, and established the Office of Dietary Supplements at the National Institutes of Health (NIH) [11]. The 1995 DGA specifically stated “supplements of vitamins, minerals, or fiber may help to meet special nutritional needs” and caveats that “‘regular use in large amounts may be harmful” and “supplements do not supply all of the nutrients and other substances present in foods” [12]. The 2000 DGA expanded beyond a focus on food variety (to reduce nutritional deficiencies) and moderation (to reduce risk of nutrition-related chronic disease) to include physical activity [13]. The 2005 DGAC adopted a more formalized systematic literature search and the focus of DGA policy shifted from the general public to policymakers, health care professionals, nutritionists and nutrition educators [14]. In 2008, the USDA established the Nutrition Evidence Library to conduct food- and nutrition-related systematic reviews and support subsequent DGAC [15]. While the 2010 DGA focused on the status of Americans’ health and the prevalence of dietary nutrient inadequacies and deficiencies, the 2015 DGA placed a greater focus on eating patterns. [16]. The 2015 DGA recognized that dietary supplements may be useful in providing one or more nutrients that otherwise may be consumed in less-than-recommended amounts, such as vitamin D [16]. The 2020 DGAC Report reiterated previous DGA reports identifying nutrients of public health concern and related biochemical or chemical indicators, acknowledging that vitamin and mineral supplements can help address vitamin and mineral intake shortfalls (Table 1), and discussing potential risks of overconsumption.

For the first time, the DGAC Report acknowledged that some terms, i.e., “essential nutrients”, “nutrients of concern” (and subgroups “under-consumed”, “over-consumed”, and “shortfall”), and “nutrients of public health concern” are consistently defined and used in the literature, and that scientific assessment and policy development are hampered by a lack of consistent use of terms for biochemical indicators such as “deficiency”, “insufficiency/inadequacy”, “sufficient/adequate” and “optimal nutrition” [17]. As consumer interest in personalized nutrition grows, the NIH has issued a notice of Intent to Publish a funding opportunity for “Nutrition for Precision Health” for research to provide more targeted and dynamic nutritional recommendations for individuals and their health care providers in January 2021 [18]. The 2020 DGA acknowledges that personal preferences, cultural traditions and budgetary considerations affect dietary choices and encourages people to choose healthy dietary patterns, recognizing that in some cases, fortified foods and dietary supplements are useful when it is not possible to meet needs for one or more nutrients, e.g., during pregnancy [2].

## 3. Understanding Population Derivation and Use of RDAs

With the exception of the discovery of mineral element essentiality in the 19th century [19], the nutritional essentiality of amino acids [20], fatty acids [21] and vitamins [22] was established early in the 20th century. Recommended Dietary Allowances (RDAs) were introduced to serve as a guide for planning adequate nutrition for the military and civilians [23] and evolved into Dietary Reference Intakes (DRIs) that consisted of Estimated Average Requirements (EAR), RDAs, Adequate Intake (AIs) and Tolerable Upper Intake Levels (UL) in 1993 [24]. RDAs are target intake levels of essential nutrients judged to be adequate to meet the needs of practically all (97–98%) healthy persons [25]. RDAs are based on: (1) studies of people eating diets that were low or deficient in the nutrient, (2) nutrient balance studies, (3) biochemical measures of tissue saturation or function, (4) nutrient intake data, (5) epidemiological observations of nutrient intake, and (6) extrapolation from animal experiments [25]. The EAR is the nutrient intake value estimated to meet the requirement defined by a specified indicator of adequacy in 50% of the individuals in a life stage and sex group [26,27,28,29,30]. An AI, a value based on observed or experimentally determined approximations of nutrient intake of healthy people, is set instead of an RDA when the scientific evidence is insufficient to calculate an EAR. The UL is the highest level of nutrient intake that is likely to pose no risk of adverse effects for most people. The DRIs do not include dietary bioactive compounds, i.e., natural constituents of food that provide health benefits [31], typically because there has been a lack of nutrient databases and dietary intake data.

## 4. Nutrient Gaps in the US and the Role of Dietary Supplements

For the past 40 years, data consistently show that Americans have not been and are still not consuming recommended amounts of whole grains, vegetables, fruits, and dairy foods, and to a lesser extent, protein food groups [17]. This translates to significant proportions of the US population who are consuming less than the EAR or AI for essential nutrients even though ~50% of US adults take at least one dietary supplement [32,33,34,35].

Products containing vitamins and minerals are the most often consumed dietary supplements [35,36]. Thus, dietary supplement use is associated with higher vitamin and mineral intake and a lower proportion of the population consuming <EAR for key micronutrients [33,37,38,39]. The decrease in % of population <EAR observed with vitamin and mineral supplement use depends on the nutrient, age, and sex groups, and other characteristics, e.g., food security, pregnancy. For example, 91.5% of men and 98.4% of women ≥19 years do not meet requirements for vitamin D from food and beverages and this shortfall drops to 66.4% of men and 59.1% of women when intake from supplements is included [40]. While dietary supplements may help meet dietary recommendations, use of multiple nutrient-containing supplements, especially high-dose forms, may increase the risk of exceeding the UL as ~ 70% and 30% of adults ≥60 years report using ≥1 and ≥4 dietary supplement in the past 30 days, respectively [35]. However, data from NHANES 2009–2012 find ≤2.6% of adults ≥71 years exceed the UL when food and supplement intake is combined, except for zinc where 5.2% exceed the UL [39]. It is also important to note that global data show micronutrient intakes do not meet dietary recommendations in many countries [41].

## 5. Defining Optimal Nutrition and Lessons from Nutrients Considered Essential

Inadequate intake of essential nutrients is known to cause deficiency diseases and increase the risk of NCD [42,43]. Albert Szent-Gyorgi, recipient of the 1937 Nobel Prize in Physiology and credited with first isolating vitamin C, is quoted as saying “The medical profession itself took a very narrow and wrong view. Lack of ascorbic acid caused scurvy, so if there is no scurvy there was not a lack of ascorbic acid. Nothing could be clearer than this. The only trouble was that scurvy is not a first symptom of a lack but a final collapse, a premortal syndrome and there is a wide gap between scurvy and full health” [44]. In other words, setting nutrient intakes using biomarker cutoffs to prevent deficiency disease is not optimizing nutritional status for health or quality of life. When nutrient biomarkers or surrogate endpoints are measured, the health gap between ‘deficient’ and ‘sufficient/adequate’ becomes apparent. Nutritional biomarkers or surrogate endpoints should reflect long term nutritional status with limited within-day or day-to-day variability. In some cases, biomarkers or surrogate endpoints may need to be adjusted for other indicators, e.g., according to WHO guidelines on the use of ferritin concentrations to assess iron status in individuals with inflammation or infection [45]. Keeping in mind that RDAs are the average dietary intake level that is sufficient to meet the nutrient requirement of 97–98% of healthy individuals and AIs are set based on experimentally derived intake levels or approximations of observed mean nutrient intakes of healthy people, we recommend that RDAs and AIs for micronutrients and bioactive compounds be established (EPA + DHA, lutein) or updated (vitamin K, magnesium) based on biomarker or surrogate endpoint concentration ranges that optimize healthy cell/organ/tissue function, as has been carried out for vitamin D.

In 1997, an AI was set for vitamin D with a goal of maintaining (in individuals with limited or uncertain sun exposure and stores) serum 25(OH)D concentrations above a defined amount to prevent vitamin D deficiency rickets or osteomalacia [46]. Recognizing that serum 25(OH)D served as a reflection of total vitamin D exposure (dietary and skin synthesis by sunlight), new RDAs were established in 2011 for healthy individuals to maintain a serum 25(OH)D concentration ~50 nmol/L (20 ng/mL), which is needed to support skeletal health [26,47]. The release of a standard reference material for vitamin D in 2009 [48] had been a significant development for vitamin D research since its availability increased the reliability of 25(OH)D data. Thus, the 2011 RDA for vitamin D was based on a shift in target serum 25(OH)D cutoffs associated with deficiency in a serum 25(OH)D to a level sufficient to maintain skeletal health in healthy people (Figure 1) [26,49]. The 2011 DRI Report states serum 25(OH)D levels <30 nmol/L (12 ng/mL) increase risk of skeletal deficiency diseases and some persons may be at risk of inadequacy between 30 and 50 nmol/L (12 and 20 ng/mL). Practically everyone would be sufficient at 50 nmol/L (20 ng/mL) whereas levels >75 nmol/L (30 ng/mL) were not consistently associated with increased benefit and there may be some concern at 25(OH)D concentrations >125 nmol/L (50 ng/mL). Using NHANES (2001–2006) data from adolescents (12–19 y), bone mineral density was positively associated with serum 25(OH)D with an inflection point at ~60 nmol/L (24 ng/mL) [50]. A systematic literature search from May 15 to December 20 2020 found low serum 25(OH)D level was significantly associated with a higher risk of COVID-19 infection [51].

Vitamin C is required to prevent scurvy; deficiency is defined as plasma ascorbic acid <11 µmol/L, while plasma saturation occurs ~70 µmol/L [28]. Scurvy can be prevented with as little as 10 mg vitamin C per day [52]. Vitamin C intake recommendations globally vary almost 3-fold based on the choice of biomarker, e.g., plasma level, tissue saturation, neutrophil ascorbate concentration, and/or a combination, used by different expert committees [53]. Vitamin C, an enzyme cofactor for collagen and carnitine biosynthesis, is essential for skeletal muscle structure and function [54]. When healthy adult men with vitamin C levels <50 µmol/L are supplemented, plasma levels increase to >70 µmol/L within a week and after 4 weeks, there is a significant increase in neutrophil vitamin C content and 20% increase in neutrophil chemotaxis post-intervention. Mortality increases with low vitamin C status in US men, but not women, with a 62% increased risk of dying from cancer with serum ascorbate levels <28 µmol/L vs. ≥74 µmol/L [55]. A meta-analysis of 15 prospective cohorts (*n* = 320,548 participants) and 3 prospective within interventional studies (*n* = 17,974 cases) finds a U-shaped association between circulating ascorbate concentrations and risk of CVD mortality (Figure 2) [56]. Even though vitamin C deficiency (<11 µmol/L) appears globally (8% in US, 12% in Singapore, 14% in Canada, 14% in France, 20% in Scotland) [57], there is still no consensus on definitions for “inadequate” or “adequate/sufficient” status [57].

Vitamins, minerals, and some amino acids and long-chain fatty acids are defined as essential nutrients because of known deficiency diseases. However, systemic availability of nutrients above cutoffs used to define deficiency may be insufficient to maintain normal cellular structure and/or function of organ systems.

## 6. Establishing Lutein and EPA + DHA DRIs and Updating Vitamin K and Mg DRIs

Some nutrients are essential to prevent deficiency diseases and for normal growth and maintenance, e.g., intrauterine growth, childhood development, etc., whereas bioactive compounds in food that are not deemed essential may still help maintain normal cellular structure and function. The Office of Dietary Supplements at the NIH defines bioactive compounds as constituents of foods or dietary supplements, other than those needed to meet basic nutritional needs, which are responsible for changes in health status; however, currently, dietary bioactive ingredients have almost no role in public policy [58]. A scientific framework has been proposed [59,60,61] requiring a safety evaluation for every ingredient before establishing recommended intakes and a UL.

The RDAs for Vitamin D and calcium, updated in 2011, were the first nutrients to have DRIs updated with a goal beyond a straightforward prevention of deficiency, i.e., an intended goal to maintain the health of a tissue/organ, i.e., skeletal health [26]. Subsequently, the US and Canadian governments sought nominations in 2013 for nutrients that should undergo the DRI process. Sixteen nutrients were nominated: arachidonic acid, choline, chromium, docosahexaenoic acid (DHA), eicosapentaenoic acid (EPA), fiber (specifically viscous fibers and fermentable fibers), magnesium, niacin, potassium, protein, saturated fat, sodium, stearic acid, vitamin B6, vitamin E, and zinc [62]. The prioritized nutrients were sodium, omega-3 fatty acids, vitamin E, and magnesium [63]. DRIs for sodium and potassium have been updated [64] and the scientific evidence for a riboflavin DRI is being scanned [65]. Arising from two conference reports, nine criteria have been identified for a food ingredient that has not been defined as a nutrient, i.e., a dietary bioactive, to qualify for DRI evaluation (Table 2) [60,61].

This report will now discuss two dietary bioactive compounds and two nutrients as case studies deserving of consideration for DRIs. Lutein was selected because, as a bioactive compound, it fulfills all nine criteria in Table 2 [66]. The second bioactive compounds, the omega-3 fatty acids EPA + DHA, and magnesium, were selected because they were prioritized by the Joint Canada–US Dietary Reference Intakes Working Group [63]. Vitamin K was chosen based on new data and requests to review vitamin K dietary recommendations [67,68]

## 7. Lutein

Lutein, lycopene, zeaxanthin, β-cryptoxanthin, and α-carotene were excluded from DRI consideration in the late 1990s because of a lack of (1) comprehensive food composition data, (2) population-based dietary intake data, (3) limited information on absorption and metabolism, and (4) insufficient data on biological actions [28,69]. Since then, lutein has been proposed for DRI review [66].

Lutein is a chemically defined xanthophyll, a class of oxygen-containing carotenoids commonly found in nature [70] with a standard reference material [71] and a publicly available database [72]. Lutein accumulates in the macular pigment in the retina and is the predominant carotenoid found in the human brain [73,74,75,76]. Factors known to affect carotenoid bioavailability, i.e., blood and tissue concentrations, include: (1) food-based factors, e.g., co-consumption of lipids, food processing, and molecular structure, (2) environmental factors, e.g., prescription drugs, smoking and alcohol consumption, and (3) individual physiological factors, e.g., age, body composition, hormonal fluctuations, and variation in genes associated with carotenoid absorption and metabolism [77]. The typical US lutein intake is 1–2 mg/day, well below the 10 mg lutein supplemented daily in Age-Related Eye Disease Study (AREDS) 2 [78]. There is strong evidence that up to 20 mg/day is safe and efficacious, and doses up to 40 mg/day have been used in studies ranging from 7 days to 24 months without reported adverse effects [66]. Extreme manipulations in primate lutein intake (xanthophyll-free diet) affect retinal pigment epithelial cells that play an important role in the visual cycle, i.e., modifying and recycling retinoids, photoreceptor materials, and nutrient transport from the blood to photoreceptor cells [79]. Foveal protection from blue light is absent in primates fed xanthophyll-free diets but evident after supplementation with lutein and zeaxanthin [80]. Lutein from foods or supplements increases blood levels and macular pigment optical density (MPOD) in the retina in a dose-dependent manner (Figure 3) [81,82,83].

MPOD is related to static indicators of visual performance, such as glare and contrast sensitivity, and dynamic measures of visual performance such as the critical flicker fusion threshold [84,85,86]. MPOD is also related to measures of cognitive function such as verbal fluency, memory, processing speed and accuracy [76]. MPOD was significantly associated with select auditory thresholds in young healthy adults [87]. Elevated lutein and zeaxanthin status appears to be associated with diminished risk of cataract [88]. While supplementation with 10 mg lutein and 2 mg zeaxanthin had no effect on advanced age-related macular degeneration (AMD) risk in AREDS, subgroup analysis showed a beneficial effect in patients with the lowest baseline intake of these carotenoids [89]. A recent systematic review and meta-analysis of six longitudinal cohort studies concluded that dietary intake of lutein and zeaxanthin was not significantly associated with a decrease in risk of developing early AMD but an increased intake of these carotenoids may be protective against late AMD [90]. A meta-analysis of 22 publications found a positive correlation of MPOD with measures of visual function, i.e., contrast sensitivity, photostress recovery, and glare disability [91]. A steroidogenic acute regulatory family protein, i.e., StARD3, has been identified in primates and subsequently identified as a human retinal lutein-binding protein [92]. In summary, there is evidence to support the promulgation of lutein DRI to achieve MPOD levels that are associated with healthy visual and brain function.

## 8. Eicosapentaenoic Acid (EPA) and Docosahexaenoic Acid (EPA)

The omega-3 (n-3) long-chain polyunsaturated fatty acids EPA (C20:5n-3) and DHA (C22:6n-3) can be produced endogenously by humans from α-linolenic acid (ALA; C18:3n-3) [93] and metabolized into hundreds of active forms, e.g., resolvins, leukotrienes, prostaglandins, thromboxane, poxytrins, maresins, etc. [94]. Biological mechanisms of action have been established and reviewed elsewhere [95]. The rate of biosynthesis from ALA is low and insufficient to meet the physiological demands for EPA + DHA [96,97]. EPA + DHA are structurally integrated via phospholipid molecules into surface membranes of heart, cardiovascular, brain and visual cells, affecting signaling pathways and function [98]. The primary sources of EPA + DHA are fish and shellfish with n-3 long-chain polyunsaturated fat intake varying from 0.023 to 0.435% of energy globally [99]. An official method to quantify fatty acids in foods is available [100] and publicly available databases exist [72]. US dietary intake studies estimate n-3 long chain intake, i.e., EPA + DHA + estimated EPA-equivalents, of 0.17 mg/day with >90% of the population consuming <0.5 g/day with ~6% of the population reporting n-3 fatty acid supplement use [101].

In 2005, the Institute of Medicine concluded there was insufficient evidence to establish DRIs for EPA and DHA [30]. The 2005 DGA recommended eating 2 servings of fatty fish per week to obtain omega-3 fatty acids, i.e., EPA and DHA, which is associated with a reduction in risk of mortality from cardiovascular disease, and it noted that “other sources of EPA and DHA may provide similar benefits” [102]. In addition to recommending that ~10% of the Acceptable Macronutrient Distribution Range for ALA can be consumed as EPA and/or DHA (~100 mg/d),Kris-Etherton et al. (2009) [103] called for the National Academies to establish DRIs for individual long-chain (≥20 carbons or greater) n-3 fatty acids. A technical committee of experts from the International Life Sciences Institute of North America proposed a DRI for EPA + DHA be established between 250 and 500 mg/day [104]. A review of 40 randomized controlled trials with EPA/DHA supplementation in 135,267 participants found EPA + DHA supplementation to be an effective lifestyle strategy for coronary heart disease prevention, and the protective effect probably increases with dosage, especially the use of 1000–2000 mg/day [105]. The EPA + DHA content of food products is available [106]. Digestion and absorption are understood. Colipase-dependent pancreatic lipase hydrolyzes triglyceride and phospholipids, with ethyl esters being digested by a bile salt-dependent carboxyl ester lipase that is affected by co-consumption of a fat-containing meal [107,108]. Dietary lipid structure does not seem to modify the incorporation of EPA and DHA found in blood [109,110]. EPA + DHA are transported into RBC and cardiac tissue at similar rates [111] and have half-life estimates in humans of 1, 67 and 22 hours for ALA, EPA, and DHA, respectively [112]. Long-chain n-3 fatty acids are sequestered in brain [113,114,115], eye [116], and adipose [117].

EPA + DHA content of RBC, the Omega-3 Index, reflects long-term intake of EPA + DHA [118] and is inversely associated with risk of CHD [119,120]. The concentration of EPA + DHA in RBC can be accurately estimated from the fatty acid composition of other blood fractions [121,122]. Best practices for the design, laboratory analysis and reporting of clinical trials involving fatty acids have been published [123]. As EPA + DHA intake increases, blood EPA + DHA concentrations increase in a dose-dependent manner [124,125]. A meta-analysis of 21 randomized clinical trials (RCT) involving high-dose EPA + DHA prescription drugs found doses of 1.8–4 g/day for intervals ranging from 8 to 261 weeks to be safe and well-tolerated [126]. DHA supplementation, alone or in combination with EPA, is associated with improved episodic memory in adults with mild memory complaints [127]. Higher plasma EPA and DHA status are associated with lower total mortality, especially CHD death, in older adults (Figure 4) [128,129].

A meta-analysis of omega-3 supplementation trials reported an 8% reduced risk of myocardial infarction and CHD death [130] but not all RCTs or prospective cohort studies have shown a consistent response [131,132]. In an extensive Cochrane review of the effects of omega-3 fatty acid supplementation (without consideration of blood EPA + DHA levels) on cardiovascular health, the authors reported moderate- and low-certainty evidence that increasing long-chain omega-3s slightly reduces risk of coronary heart disease mortality and events, and reduces serum triglycerides (evidence mainly from supplement trials) [133]. Two meta-analyses report reduced relative risk per 1 standard deviation increase in blood fatty acid level for CHD: EPA + DHA (0.75; 95% CI: 0.62–0.89) [134] and EPA (0.91; 95% CI: 0.82–1.00) [135]. Indeed, the International Society for the Study of Fatty Acids and Lipids has recommended that all research studies include measurement of n-3 fatty acids at baseline and follow-up [136]. In a prospective cohort study with 1625 deaths (total, CVD, and CHD), collected between 1992 and 2008 and a total of 30,929 person-years, individuals in the highest plasma EPA + DHA quintile lived an average of 2.22 more years after age 65 years than did those in the lowest quintile [128]. The Women’s Health Initiative Memory Study, a prospective cohort, found an 8% reduction in risk of death with higher blood EPA + DHA levels [137]. A meta-analysis of prospective observational studies found that individuals with an Omega-3 Index >8% were at 35% lower risk for death from any cause than those with an Omega-3 Index <4% [138]. Circulating DHA concentrations were significantly lower in individuals with mild cognitive impairment relative to controls [139]. Higher blood EPA + DHA levels appear to protect people exposed to ambient particulate matter air pollutants as they have muted blood fibrinogen responses [140] and greater brain volumes [141]. In conclusion, the growth in evidence associating higher EPA + DHA levels with beneficial health outcomes coupled with updated safety data is sufficient to justify setting EPA + DHA DRIs to achieve target blood ranges.

## 9. Vitamin K

Vitamin K is a fat-soluble vitamin that exists naturally in multiple forms: 1) vitamin K_1_ consisting of a phylloquinone with a 2-methyl-1,4-napthoquinone ring with a phytyl group at the 3-position, and 2) vitamin K_2_, or menaquinone (MK) forms where the phytyl group is replaced with 4–10 repeating isoprenoid units, MK-4 through MK-10, respectively [142,143]. Vitamins K_1_ and K_2_ were isolated in 1939 [144]. Vitamin K_1_ is found in green leafy vegetables and vegetables oils [72], MK-4 through MK-6 are present in low levels in animal based foods, e.g., some cheeses and chicken meat, and MK-7 is found in fermented soybeans (natto) where it is formed by bacteria during fermentation [143]. Vitamin K is also produced by gut microbiota but their contribution to vitamin K status is unclear [68]. In 1935, vitamin K was identified as an antihemorrhagic factor and a convenient analytical method for food was published in 1936 [145]. All forms of vitamin K serve as a cofactor for posttranslational carboxylation of specific protein-bound glutamyl residues to γ-carboxyglutamate (Gla) that are essential for the formation of several coagulation factors (II, VII, IX and X) and inhibitors (proteins C and S) in the liver [146,147]. Gla proteins not related to blood clotting are osteocalcin (OC, synthesized in bone) and matrix Gla protein (MGP, primarily synthesized in cartilage and the vessel wall) [148]. Low vitamin K intake is associated with low bone mineral density increased fracture risk, and increased risk of CVD and mortality [148]. Higher levels of under-carboxylated osteocalcin (*u*cOC) are a marker of hip fracture in elderly women [149]. Supplementation with vitamin K_2_ (375 µg MK-7/day) decreased *uc*OC after 3 months and preserved trabecular bone structure at the tibia at 12 months [150]. Based primarily on indicators of coagulation and dietary phylloquinone (K_1_) intake, AIs for vitamin K were established [27]. Vitamin K consumption from food or supplements is not associated with adverse effects, including toxicity, in humans or animals with the caveat that there was insufficient high vitamin K intake data in humans to establish a UL (highest recorded intake was 367 µg/day) [27]. Since the DRIs were issued, a protective role for vitamin K, specifically vitamin K_2_, in bone health has emerged [151,152] and it has been noted that the AI may be insufficient for full carboxylation of all vitamin K-dependent proteins [143]. The essential role of MGP in inhibiting arterial calcification was confirmed in rats treated with warfarin to induce rapid arterial calcification [153]. Using this rat model, increasing dietary vitamin K intake increases vitamin K concentrations in the aorta and blunted cardiovascular calcification [153,154]. At the tissue level, vitamin K_1_ is converted to MK-4 [155] with vitamin K_2_ being effective at lower doses than K_1_ [156]. Supplementation with vitamin K has been shown to block age-related arterial stiffening in postmenopausal women [157] and retard postmenopausal bone loss [158,159]. Often prescribed to prevent thomboembolisms, vitamin K antagonists interfere with γ-carboxylation of Gla-proteins in mice [160]. Vascular calcification is a predictor of cardiovascular mortality and *u*cOC levels and MK-7 supplementation induces a time- and dose-dependent reduction in circulating *u*cOC, dephospho-uncarboxylated matrix Gla protein (dp-*u*cMGP) levels in hemodialysis patients [161]. A meta-analysis of 19 RCTs with postmenopausal women with or without osteoporosis found that vitamin K_2_ supplementation decreased *u*cOC and increased OC, indicating a positive effect on bone metabolism and reduced the incidence of fractures with a risk ratio of 0.63 [162]. The authors concluded vitamin K_2_ supplementation was effective for maintaining vertebral and forearm bone mineral density (BMD) in postmenopausal women with osteoporosis but there was no significant effect in postmenopausal women without osteoporosis. Calcification of the coronary artery has been identified as a marker of increased CVD risk in humans [163,164,165]. In a double-blind RCT, MK-7 supplementation of healthy, prepubertal children resulted in increased blood MK-7 concentrations and OC (vs. controls) but bone markers and coagulation parameters did not differ between treatments [166]. In an RCT with 244 postmenopausal women using 180 µg MK-7 per day for 3 years, vitamin K2 supplementation decreased dp-*u*cMGP values by 50% (vs. controls) and significantly improved vascular stiffness indicators [157]. Subsequently, MK-7 supplementation has been reported to increase circulating c-OC and *u*cOC levels [167]. In a meta-analysis of 13 controlled RCTs and 14 longitudinal trials, vitamin K supplementation was associated with a 9% reduction in vascular calcification, 44% reduction in dp-*u*cMGP, and 12% reduction in *u*cOC, all indicators pointing to a reduction in vascular disease and CVD mortality (Figure 5) [168].

A meta-analysis of 11 prospective cohort studies concluded that high blood dp-*u*cMGP level, an indicator of vitamin K insufficiency, is an independent predictor of cardiovascular disease and mortality [169]. In conclusion, recent insights into the metabolism of vitamin K_1_ and K_2_ forms and their metabolites as biomarkers of disease risk and new evidence linking dietary vitamin K in food or supplement with maintaining normal bone, blood clotting, and cardiovascular function justify a systematic review of the scientific literature, reevaluation of the vitamin K AI, and possibly dividing the DRI into vitamin K_1_ and K_2_ forms [67].

## 10. Magnesium (Mg)

Mg, a required cofactor for over 600 enzyme reactions and 50% of total body Mg content found in bone [170], has an EAR derived from intake data and limited balance studies [46]. Apatite-bound Mg in bone cannot be mobilized even under extreme depletion, whereas Mg absorbed to the surface of mineral crystals can be mobilized during hypomagnesemia [171]. Most men and women are not consuming the Mg EAR from food alone, i.e., green vegetables, nuts, seeds, dried beans, whole grains, and meats, and supplements containing Mg contribute importantly to total dietary intake [172,173]. Mg has been repeatedly identified as an under-consumed nutrient [16,17,102,174] with a greater percentage of non-Hispanic Black people <EAR for Mg, calcium, and phosphorus than non-Hispanic white people across all ages [175]. Older obese adults have a greater risk of inadequate Mg intake than their healthy-weight counterparts [176]. Thirty percent of Mexican-American and non-Hispanic Black women have dietary Mg intake <EAR with the percentage varying by body weight status [177]. Mg is generally absorbed from the gut as an ion through transcellular and paracellular pathways [178]. Due to the importance of passive paracellular Mg^2+^ absorption, the amount of Mg in the gut is the major factor controlling absorption [178]. Other factors affecting Mg requirement include body mass, obesity, background diet (calcium, type of fiber, vitamin E and selenium), and oxidative stress [179]. Mg is essential in the metabolism of vitamin D as low dietary Mg intake can alter vitamin D–parathyroid hormone balance [180] and large doses of vitamin D can deplete Mg [181]. Evidence from 27 different balance studies in 243 healthy individuals found age and sex do not appear to affect urinary Mg excretion [182]. Some think the 1997 DRIs were set too high and lower levels have been proposed [182], whereas others think the proposed levels did not consider numerous physiological factors such as total body weight [179,183].

Cellular Mg levels are strictly regulated and eight cation channels have been identified with transient receptor potential melastatin (TRPM7) being the most selective channel for Mg in the heart, blood vessels, lungs, liver, brain, intestine and spleen, whereas TRMP6 is mainly responsible for regulating total body magnesium level via the kidney and intestines [170]. The serum total Mg concentration (STMC) normal reference range used clinically is 0.85–0.96 mmol/L (1.70–1.92 mEq/L or 2.06–2.33 mg/dL) [184] and is derived from values measured in a healthy population (NHANES I) [185] rather than the relationship between serum Mg and clinical outcomes which can manifest <0.85 mmol/L [186]. While overt Mg deficiency is not common, almost every organ system is affected by the availability of Mg [186,187]. Strong correlations of Mg concentration with increased risk of several chronic diseases indicate that Mg status should be assessed more routinely [188], not only to reassess DRI chronic disease endpoints [189] but also for optimization of nutritional status and body stores.

Using baseline STMC data from 14,353 participants (NHANES 1 1971–1975) with a median follow-up of 28.6 y (until 2011), very low STMC (<0.7 mmol/L) was significantly associated with a 34% increased risk of all-cause mortality (HR = 1.34; 1.02–1.77) (Figure 6) and trended significance for cancer (HR = 1.39; 0.83–2.32), CVD (HR = 1.28; 0.81–2.02) and stroke (HR = 2.55; 1.18–5.48) [190]. A cross-sectional, population-based survey (2012–2013) of 5561 participants living in Canada found 9.5–16.6% of adults and 15.8–21.8% of adolescents surveyed had a STMC <0.75 nmol/L and STMC was negatively associated with diabetes, BMI, serum glucose, serum insulin, HbA_1C_, and HOMA-IR [191]. Having diabetes was associated with 0.04–0.07 mmol/L lower STMC compared to not having diabetes [191].

In summary, in addition to the established evidence linking Mg with healthy bones, there is emerging evidence of a relationship between STMC and risk of NCD. Current DRIs need to be updated to consider physiological factors, i.e., body mass and obesity, and assess the interaction of other dietary factors, such as the amount of calcium, protein, dietary fiber, antioxidants, vitamin E and selenium. This will require updating food databases [72] to maintain accurate estimates of Mg, sodium, potassium and calcium content in food. The lack of a standardized biomarker and STMC reference range with appropriate cutoffs to accurately assess Mg status remains a challenge. With only 0.3% of total body Mg in serum, the combination of STMC, urinary Mg excretion and dietary Mg history adjusted for body weight may be the best means to assess Mg status [186].

## 11. Personalizing Nutrition Guidance and Improving Population-Based Assessment

Precision health depends upon the integration of dietary history, supplement use, demographics, behavior, lifestyle, social, cultural, economic, occupational, and environmental factors. It also encompasses personal information, i.e., age, sex, nutrition-related biomarkers, genetics, microbiome, etc. The DRIs provide a set of reference values that are useful in planning and assessing the adequacy of nutrient intakes of healthy individuals. Nutrient intake assessments provide insight into individual dietary patterns and inform nutritional advice, i.e., personalizing guidance, but it can take days of intake data to predict an individual’s usual intake, even months for nutrients such as vitamin A [192]. Dietary intake records may still not reflect nutritional status because of reporting inaccuracies, outdated and/or incomplete food and supplement databases because of the rapid evolution of commodity, food and supplement offerings, and a lack of bioavailability information. Technology is streamlining dietary data collection, analysis and interpretation and diagnostic devices measuring validated biomarkers and metabolic profiles can help assess the availability of essential nutrients and dietary bioactive components to cells, tissues and organs [193]. The proliferation of smartphone dietary apps, direct-to-consumer sale of personalized formulations with re-order reminders, wearable devices, and submission of biological samples raises ethical and privacy considerations. Nevertheless, personalized or precision nutrition approaches will benefit from the validation of biomarkers and metabolic health, standardization of methods of analysis, consensus on reference ranges defining deficient, insufficient, adequate, and optimal status with consideration for an individual’s sex, age, life stage, nutrigenomics, microbiomics, and the algorithms that bring these factors together [194].

With objective measures of nutritional status, e.g., RBC folate, uncarboxylated vitamin K-dependent proteins, or serum 25(OH)D concentrations, researchers can characterize relationships between objective, biochemical measures of nutritional status and functional outcomes to define nutrient ranges for vitamins, minerals and bioactive compounds that support healthy cellular, organ, and tissue function, i.e., optimize health. The greatest risk of false-negative or false-positive tests is not in the optimal range; risk is greatest at the tails of the distribution surrounding the cutpoints being used to diagnose a deficiency disease or nutritional excess. Finally, by understanding the nature of nutritional status and functional outcome relationships, e.g., 25(OH)D and skeletal health, it is possible to define optimal ranges and use these data to update or establish DRIs for the general healthy public (Figure 1). The adoption of point-of-care diagnostic tools by health care professionals will enable accurate assessment of the nutritional status of individuals, personalization of dietary guidance, and follow-up to determine if a deficiency, insufficiency, or excess has been remediated. These same biomarker or surrogate endpoint measures can be used by public health professionals to accurately assess nutrition interventions [195]. Point-of-care nutrition monitoring diagnostic tools are often invasive, i.e., finger stick, but advances in wearable and saliva-based technologies are likely. While the adoption of point-of-care diagnostic technologies may be more costly than measuring dietary intake, manufacturing economies of scale will decrease cost and increase availability. Most importantly, by identifying individuals and communities at risk of undernutrition and overnutrition, the risk of nutrition-related communicable diseases may be reduced.

## 12. Conclusions

Underconsumption of some essential nutrients and food bioactive components, especially from food alone, is still a concern in the US population, even though dietary guidance recognizes the contributions from food fortification and vitamin and mineral dietary supplements. The DGA aim to provide recommendations for healthy eating to promote health and prevent disease, and are updated periodically to incorporate current scientific evidence, yet recommended intakes still rely on DRIs established decades ago. Moreover, US DRIs define dietary intakes needed to maintain nutritional status at levels that prevent vitamin/mineral deficiency diseases; DRIs should be revised to be based on intake levels that provide cells, organs and tissues with access to adequate amounts of micronutrients (and bioactives) to function optimally, i.e., healthy structure/function outcomes. It is recommended that DRIs for vitamin K and Mg be updated and DRIs be established for lutein and EPA + DHA. Precision or personalized nutrition offers unprecedented opportunities to assess the nutritional status of healthy individuals and to personalize individual dietary guidance. The field will be further advanced by investments in the research, development, evaluation and validation of innovative algorithms and technologies to better assess dietary intake from food and dietary supplements, nutritional status, metabolomic fingerprints, microbiome profiles, and metagenomic measures across age, sex, and physiological (growth, pregnancy, lactation) classifications. Continual updating of food and supplement composition databases will be required. The adoption of diagnostic devices, including the measurement of nutritional biomarkers, is a foundation of precision health. Objective measures of nutritional status, e.g., RBC folate, uncarboxylated vitamin K-dependent proteins, or serum 25(OH)D concentrations, will allow for the evaluation of interventions ranging from mandatory food enrichment and fortification to the distribution and use of dietary supplements on health outcomes. The data generated via precision nutrition on nutritional status and physiological function will be important to guide public health policy on federal nutrition programs.

## Figures and Tables

**Figure 1 nutrients-13-01844-f001:**
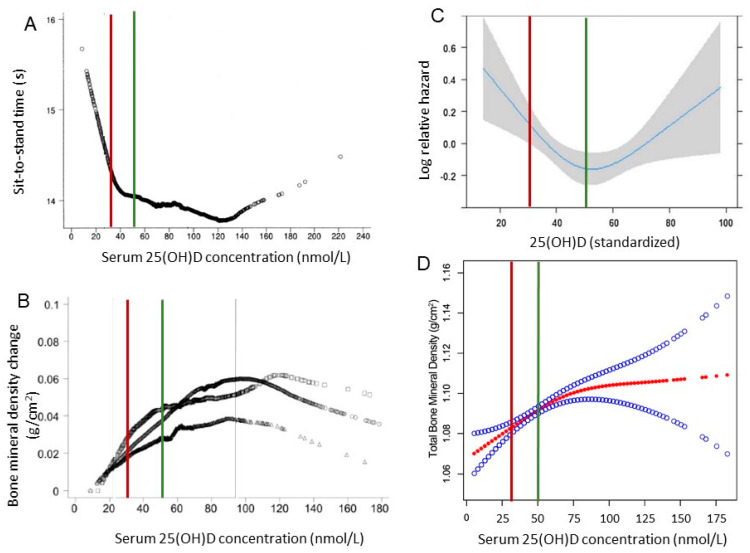
Serum 25(OH)D concentration relationship with different structural or functional outcomes. (**A**). Serum 25(OH)D concentrations relationship with sit-to-stand time in adults with mean age 71 years (49% female) controlled for sex, age (5-year categories), race/ethnicity, BMI, poverty income ratio, daily calcium intake, number of medical comorbidities, use of a walking device, self-reported arthritis, activity level, and month of vitamin D measurement. From [47]. (**B**). Serum 25(OH)D concentration relationship with bone mineral density in adults ≥50 years after adjustment for sex, age, BMI, smoking, calcium intake, estrogen use, month of vitamin D measurement, and poverty income ratio. From [47]. (**C**). Spline curve describing the association between 25(OH)D concentration and recurrent fallers in the total population. From [49]. (**D**). The association between serum 25(OH)D level and total bone mineral density from NHANES 2001–2006 among 5990 adolescents (12–19 years). Solid red line represents the smooth curve fit between variables. Blue band represents the 95% confidence interval from the fit. Adjusted for age, gender, race/ethnicity, income to poverty ratio, education, physical activity, body mass index, calcium use. From [50]. Vertical line indicates vitamin D deficiency cutoff, i.e., serum 25(OH)D < 30 nmol/L (red), used in setting DRI [46] and insufficiency cutoff, i.e., serum 25(OH)D < 50 nmol/L (green), used in updating DRI [26].

**Figure 2 nutrients-13-01844-f002:**
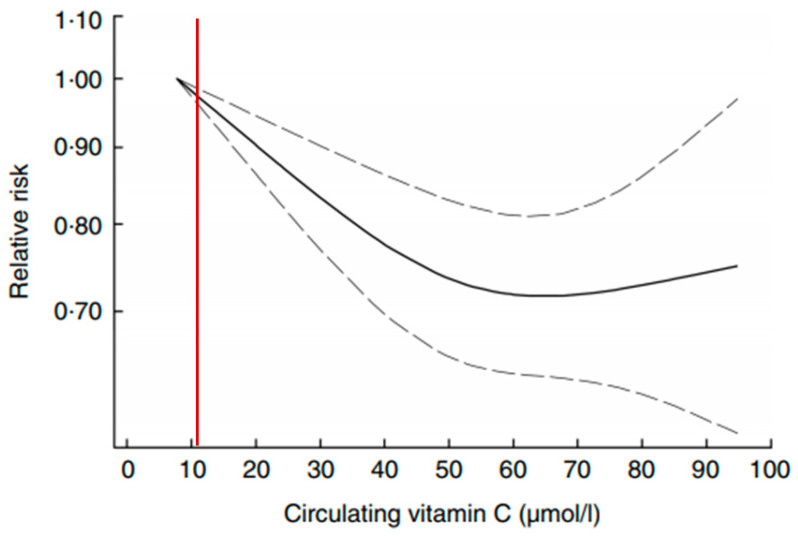
Dose–response association between vitamin C concentration and risk of total CVD mortality with 95% CI from 6 studies with 45,040 participants and 2992 cases. From [56]. Vertical red line indicates vitamin C deficiency, i.e., blood ascorbic acid < 11.4 µmol/L, used in setting DRI [28].

**Figure 3 nutrients-13-01844-f003:**
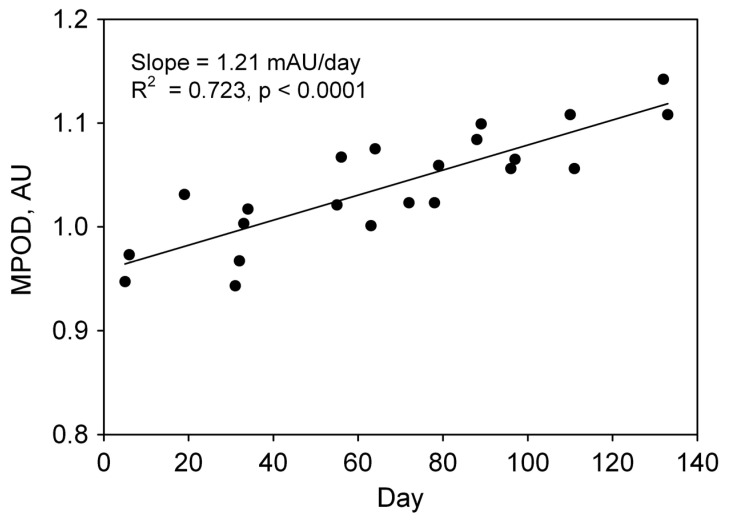
Serum lutein and macular pigment optical density (MPOD). Rate of change of MPOD (milli-Absorbance units) as a function of fractional change in serum lutein concentration (plateau value minus pre-supplementation value, divided by pre-supplementation value). Data from 46 healthy subjects randomly assigned to 0, 5, 10, and 20 mg of free, unesterified lutein in soft-shell gelatin capsules and whose standard deviations in the rates of change of MPOD were <0.250. From [83].

**Figure 4 nutrients-13-01844-f004:**
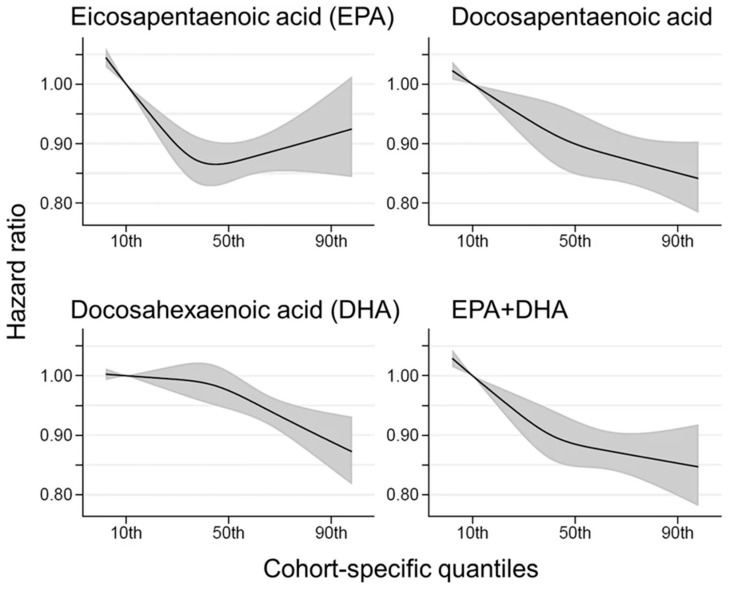
Multivariate-adjusted relationship of blood long-chain, omega-3 fatty acids with all-cause mortality. De novo pooled analysis using data from 17 prospective cohort studies with a median 16 y follow-up, 15,720 deaths among 42,466 participants, and an average baseline age of 65 years. The solid lines and shaded area represent the best estimates and 95% CI, respectively. The 10th percentile was selected as a reference level (HR = 1) and the x-axis depicts 5th to 95th percentiles. Potential nonlinearity was identified for EPA (*p* = 0.0004) but not for others (*p* > 0.05). All HRs are adjusted for age, sex, race, field center, body-mass index, education, occupation, marital status, smoking, physical activity, alcohol intake, prevalent diabetes, hypertension, and dyslipidemia, self-reported general health, and the sum of circulating 6 polyunsaturated fatty acids (linoleic plus arachidonic acids). From [129].

**Figure 5 nutrients-13-01844-f005:**
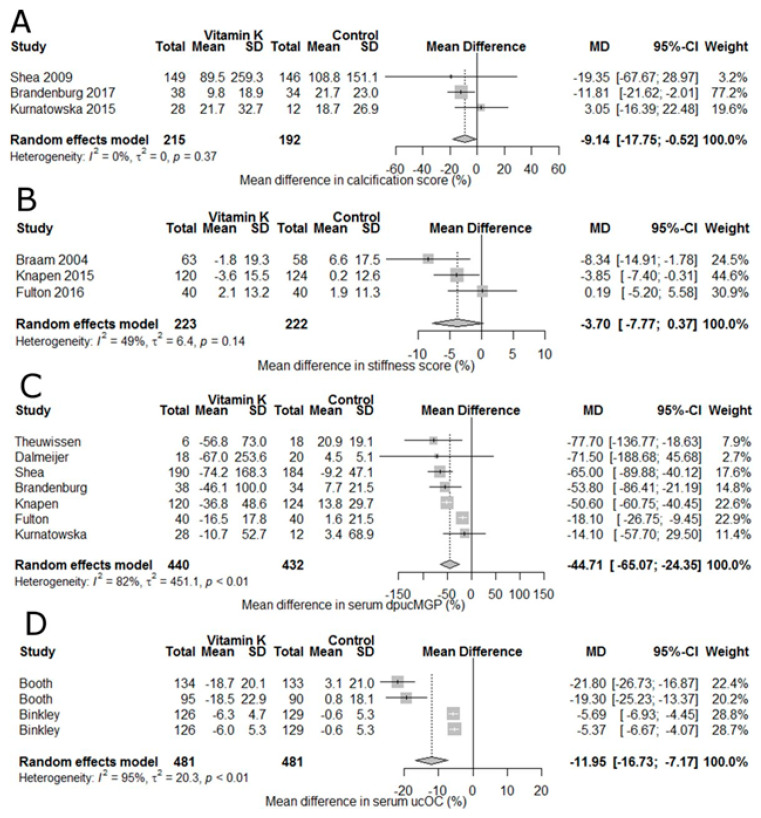
Forest plots showing the effect of vitamin K supplementation。 (**A**) % change in vascular calcification. (**B**) Vascular stiffness. (**C**) Serum dephospho-uncarboxylated matrix Gla protein (dp-*u*cMGP). (**D**) serum uncarboxylated osteocalcin (*u*cOC). Data are presented as mean % difference and 95% CI. From [168].

**Figure 6 nutrients-13-01844-f006:**
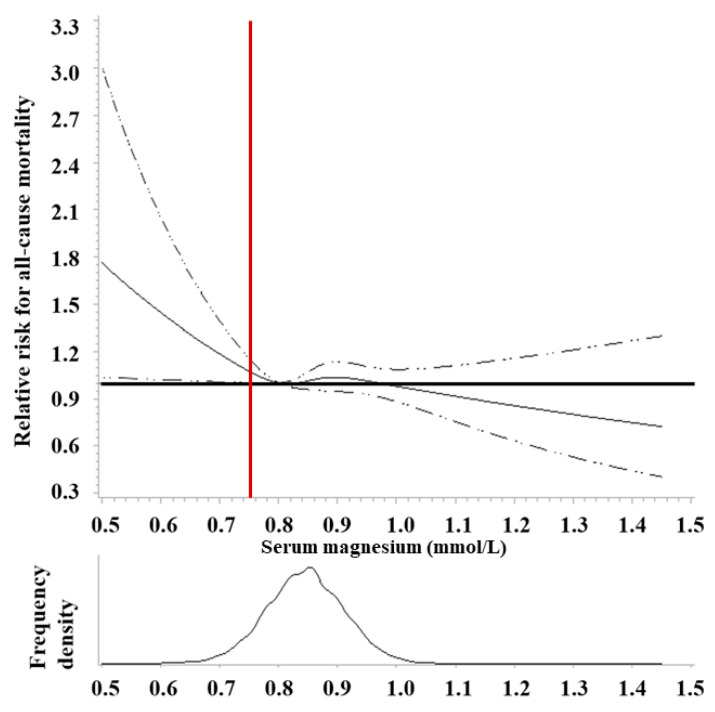
Adjusted hazard ratios of serum magnesium with all-cause mortality in the US adult population (NHANES I 1971–2011). The solid curve is HR calculated by restricted cubic splines with knots at serum Mg levels of 0.73, 0.82, 0.87, and 0.96 mmol/L, a reference HR = 1 set at 0.80 mmol/L, and adjusted using weighted Cox regression model for age, sex, race/ethnicity, education, family income, smoking, alcohol, physical activity, BMI, history of diabetes, hypertension, and vitamin and/or mineral supplement uses. From [190]. Vertical red line indicates magnesium depletion cutoff, i.e., serum Mg concentration <0.75 mmol/L, used in setting DRI [46].

**Table 1 nutrients-13-01844-t001:** Select food components (nutrients) of public health concern with summary by life stage. Adopted from Dietary Guidelines Advisory Committee (2020) [17].

Food Component	Life Stage	Dietary Intake Metric	Biochemical or Clinical Indicator	Associated Health Condition	Last DRI Review
Potassium ^1^	≥1 y, including pregnant or lactating women	% > AI	24 h urinary excretion	Hypertension and cardiovascular disease	2019
Sodium	≥1 y, including pregnant or lactating women	% > CDRR ^2^	24 h urinary excretion	Hypertension and cardiovascular disease	2019
Calcium ^1^	≥1 y, including pregnant or lactating women	% < EAR	No reliable biochemical marker exists	Impaired peak bone mass accrual; low bone mass and osteoporosis	2011
Vitamin D ^1^	≥1 y, including pregnant or lactating women	% < EAR	Serum 25(OH)D concentrations	Impaired peak bone mass accrual; low bone mass and osteoporosis	2011
Iron ^1^	Infants fed human milk; adolescent, pre-menopausal, pregnant women	% < EAR	Serum ferritin, soluble transferrin receptor, hemoglobin	Iron deficiency and iron deficiency anemia	2001
Iodine	Pregnant women	% < EAR	Urinary iodine concentrations	Impaired neurocognitive development	2001
Folic Acid	Pregnant women, 1st trimester	% < EAR	Serum and red blood cell folate	Neural tube defects	1998

^1^ FDA’s designation as a nutrient of “public health significance”. ^2^ CDRR = Chronic Disease Risk Reduction.

**Table 2 nutrients-13-01844-t002:** Criteria to qualify for DRI evaluation and inclusion in DGAs. Adapted from [61].

Criterion	Additional Information
Commonly used definition of the substance	Definition matches method of analysis
A method of analyzing the substance consistent with the definition	Preferably validated by multi-center analysis
Database of the amount of the nutrient or bioactive in food (and supplements)	Preferably global and regularly updated
Prospective Cohort studies	Both sexes and showing relationship between outcome and dietary intake, or preferably biochemical or clinical indicator
Clinical trials on digestion, absorption, transport, and excretion of the substance	Important to understand level of intake, factors affecting absorption, metabolism, and excretion
Clinical trials on efficacy and dose–response	Conducted in healthy populations with bioactive being measured along with accepted endpoint or biomarker
Safety data at anticipated level of intake	Should include data from special populations, e.g., children, pregnant or lactating women
Systematic reviews and/or meta-analyses showing efficacy	Required by IOM for setting DRI and inclusion in DGA recommendations
A plausible biological explanation for efficacy	Not required but nevertheless important

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
