# Peer review of "Beyond Nutrient Deficiency—Opportunities to Improve Nutritional Status and Promote Health Modernizing DRIs and Supplementation Recommendations"

_nutrients, 2021, doi:10.3390/nu13061844_

Round 1

Reviewer 1 Report

This article entitled “Beyond Nutrient Deficiency – Opportunities to Improve Nutritional Status and Promote Health” is an interesting review of the historical transition of policy regarding nutrition in the US. Also, the authors argue the necessity of revisions of or additions to the dietary guidelines for some nutrients (vitamin K and Mg) and bioactive compounds (lutein and EPA+DHA) based on the rigorous review. I found the paper to be overall well written, and the authors performed careful and thorough review processing. However, to be friendlier for general readers, I raise several comments listed below that should be addressed.

Major

  1. Please discuss the necessity of deficiency-oriented DRIs carefully in the present US situation. As the authors argued, cells/organs/tissues oriented DRIs would be possible future directions. On the other hand, I think the recommendation for prevention of deficiency is still important under the particular situation (e.g., disaster).

  1. The authors should emphasize that the improvement of dietary patterns and/or food choices would solve many nutritional problems, including overweight/obesity. The conclusion of the manuscript could mislead that exploring the potential risk of nutritional insufficiency and taking supplements only the way to solve the present nutritional problems among the U.S population.

  1. A discussion from the ethical aspect about personal and precision assessment of nutritional intake by innovative technology would be needed. Such a method would be better than the present measures based on dietary records regarding the accuracy but often invasive (e.g., sampling blood). Also, it is not ignoble that the cost of using such innovative methods and taking supplements. The introduction of high-cost innovative personal/precise assessment may increase nutritional health inequality among the population.

  1. Please refer to review articles or appropriate evidence regarding the relationship between an increase in vitamin K consumption and lowering the risk of bone fracture, if possible.

  1. I miss a deeper discussion about the target population of the revision of the DRIs that the authors argued. For example, even if the DRIs were revised for vitamin K, they should be adopted only for the high-risk population regarding bone fracture (e.g., post-menopausal women and older adults).

Minor

  1. Please check if the reference numbers are appropriate, in particular those at figure legends.

Author Response

1. Please discuss the necessity of deficiency-oriented DRIs carefully in the present US situation. As the authors argued, cells/organs/tissues oriented DRIs would be possible future directions. On the other hand, I think the recommendation for prevention of deficiency is still important under the particular situation (e.g., disaster).

Wording has been clarified in the abstract (39-44) and throughout to clarify the point that RDAs currently based on biomarkers or surrogate endpoint concentration that prevent deficiency in health individuals be updated (vitamin K, magnesium) or established (lutein, EPA+DHA) based on biomarker or surrogate endpoint ranges that support normal cell/organ/tissue structure and function (lines 194-204, 212-221).

2. The authors should emphasize that the improvement of dietary patterns and/or food choices would solve many nutritional problems, including overweight/obesity. The conclusion of the manuscript could mislead that exploring the potential risk of nutritional insufficiency and taking supplements only the way to solve the present nutritional problems among the U.S population.

The introduction has been modified (63-69) to clarify the basis of DGA is to guide people to choose foods and food patterns that will provide 40 different nutrients, essential vitamins, minerals, amino acids, and fatty acids and energy needs (calories) to stay healthy. Energy imbalance that contributes to overweight/obesity is clearly a nutritional and health problem. However, the focus in this paper is on identifying vitamin/mineral/bioactive biomarker or surrogate endpoint ranges associated with healthy cellular, organ, and tissue function. Using this data, then an RDA or AI can be determined to maintain health. To increase clarity for the readers, we have replaced nutrient intake (which includes calorie components) with vitamin, mineral, or bioactive compounds when referencing RDA, AI, etc. (e.g. 597-600; 624-625).

3. A discussion from the ethical aspect about personal and precision assessment of nutritional intake by innovative technology would be needed. Such a method would be better than the present measures based on dietary records regarding the accuracy but often invasive (e.g., sampling blood). Also, it is not ignoble that the cost of using such innovative methods and taking supplements. The introduction of high-cost innovative personal/precise assessment may increase nutritional health inequality among the population.

 We have added a statement to address ethical issues (lines 588-591) and cost (611-616).

3. Please refer to review articles or appropriate evidence regarding the relationship between an increase in vitamin K consumption and lowering the risk of bone fracture, if possible.

 Citation added to Ronn et al (2016) (460-461).

4. I miss a deeper discussion about the target population of the revision of the DRIs that the authors argued. For example, even if the DRIs were revised for vitamin K, they should be adopted only for the high-risk population regarding bone fracture (e.g., post-menopausal women and older adults).

The RDAs (and AIs) establish average dietary intake levels to meet the needs for 97-98% of healthy individuals. We have modified the text throughout (39-44; 198-204; 220-222; 604-606; etc) to make it clear to the reader that the emphasis is on updating or establishing DRIs for healthy individuals. The Institute of Medicine has issued guidelines for DRIs based on chronic disease but the focus of this manuscript is healthy individuals.

Minor

5. Please check if the reference numbers are appropriate, in particular those at figure legends.

Verified. Thank you.

Reviewer 2 Report

My take on this paper is that it concerns a call to revise DRI's of some nutrients (vitamin K and magnesium) and bioactives (lutein and DHA/EPA). In general, it advocates for using biochemical markers rather than functional markers in setting nutrient requirements. Moreover, authors call for finetuning of DRI's by distinguishing between deficient/insufficient/adequate/optimal intake.

The paper is generally well-written and engaging, and the topic is important. After careful reading I am left with a number of questions:

  • Why does the paper start from the DGA viewpoint, since it focuses mainly on specific nutrients/bioactives and not on foods/dietary patterns?
  • The chosen focus nutrients (lutein, DHA/EPA, vitamin K and magnesium) are presented as case studies; yet it doesn't become clear why these were chosen and how they relate to other nutrients/bioactives that may need updating/setting of DRI's. Please provide more context.
  • How do authors propose to define deficient/ insufficient/ adequate/ optimal intake for the case nutrients/bioactives?
  • Assessing nutrient status of individuals is all but straightforward, i.e. day-to-day variation, homeostatic control, inflammation. I miss a reflection on this in relation to setting DRI's based on biochemical markers.
  • Apart from using biochemical status to set DRI's, authors also  envisage the use of biochemical status in clinical practice. This would indeed strengthen the diagnosis of individual cases, but also comes with drawbacks, e.g. invasiveness, costs, false positives, false negatives, etc. I miss reflection on this.
  • No mention is made of EFSA's more recent nutrient recommendations for vitamin K (2017) and magnesium (2015), please clarify in the paper how EFSA's recommendations compare to the older US guidelines.
  • No mention is made of the importance of sun exposure for vitamin D, nor of intrinsic production of vitamin K by the microbiota. This hampers the setting of appropriate DRI's and should be explained.

Minor points:

Lines 179-183: Can you specify the AI and RDA so that it becomes clear how they compare? The section that follows focuses on maintaining status, but it does not become clear how this relates to intake. Moreover, no mention is made about the contribution of sun exposure.

Lines 193-195: This is suggestive (incomplete) and out of scope here, please remove.

Lines 259-260: The order in this sentence is nutrients and bioactives, but the following section is in reverse order. Please align.

Figure 3: The study represented by panel A used a combination of high-dosed nutrients and is not specific for lutein. Hence, it does not represent lutein intake and MPOD and should rather be removed.

Line 403-404: Note that soybeans contain only a modest amount of vitamin K as phylloquinone (K1). Natto contains menaquinone (K2) which is formed by bacteria during fermentation. It cannot be claimed that legumes in general are rich sources of MK-7.

Author Response

The paper is generally well-written and engaging, and the topic is important. After careful reading I am left with a number of questions:

  • Why does the paper start from the DGA viewpoint, since it focuses mainly on specific nutrients/bioactives and not on foods/dietary patterns?

The introduction has been modified to clarify the basis of DGA is to guide people to choose foods and food patterns that will provide 40 different nutrients, essential vitamins, minerals, amino acids, and fatty acids and energy needs (calories) to stay healthy (63-69). It also provides insight on the contributions of food, including enriched and fortified foods, and dietary supplements to nutrient intake and nutritional status.

  • The chosen focus nutrients (lutein, DHA/EPA, vitamin K and magnesium) are presented as case studies; yet it doesn't become clear why these were chosen and how they relate to other nutrients/bioactives that may need updating/setting of DRI's. Please provide more context.

Justification for these four case studies are included at the end of “Establishing Lutein and EPA+DHA and Updating Vitamin K and Mg DRIs” section (299-305).

  • How do authors propose to define deficient/ insufficient/ adequate/ optimal intake for the case nutrients/bioactives?

The definition of cut-points for deficient/insufficient/adequate/optimal intake levels or to specify biomarker or surrogate endpoint levels to specify deficiency or excess in clinical practice is beyond our scope. Instead we provide a framework based on optimal health and call for vitamin K and Mg DRIs to be updated and for DRIs to be established for lutein and EPA+DHA. We acknowledge that the challenges in establishing cutpoints for clinical diagnosis of disease (600-606).

  • Assessing nutrient status of individuals is all but straightforward, i.e. day-to-day variation, homeostatic control, inflammation. I miss a reflection on this in relation to setting DRI's based on biochemical markers.

Yes, these are important issues. We have emphasized the need for biomarkers or surrogate endpoints that reflect long term nutritional status with limited within-day or day-to-day variability (194-196), adjustment when necessary for other indicators such as inflammation (197-198), and the importance of standard reference materials (216-218) to make this clearer to readers.

  • Apart from using biochemical status to set DRI's, authors also  envisage the use of biochemical status in clinical practice. This would indeed strengthen the diagnosis of individual cases, but also comes with drawbacks, e.g. invasiveness, costs, false positives, false negatives, etc. I miss reflection on this.

Important considerations. See additions on risk of false positives/negatives in discussion (600-603), invasiveness (611-612), cost and availability (612-616).

We have tried to clarify that the purpose of

  • No mention is made of EFSA's more recent nutrient recommendations for vitamin K (2017) and magnesium (2015), please clarify in the paper how EFSA's recommendations compare to the older US guidelines.

We have added these citations in the belief that these reviews emphasize the need for other nations, including the US, to conduct their own DRI reviews (300-305).

  • No mention is made of the importance of sun exposure for vitamin D, nor of intrinsic production of vitamin K by the microbiota. This hampers the setting of appropriate DRI's and should be explained.

Modified text (213-218) to reference sun exposure for vitamin D. Added sentence on vitamin K being produced by gut microbiota (450-451).

Minor points:

Lines 179-183: Can you specify the AI and RDA so that it becomes clear how they compare? The section that follows focuses on maintaining status, but it does not become clear how this relates to intake. Moreover, no mention is made about the contribution of sun exposure.

Clarifying language has been added as requested (154-155).

Lines 193-195: This is suggestive (incomplete) and out of scope here, please remove.

The sentence has been edited to accurately reflect the remarks of the 2011 DRI Committee, i.e. “At this time, the scientific data available indicate a key role for calcium and vitamin D in skeletal health and provide a sound basis for DRIs” on page 14 of the summary (214-216). Thank you for helping get the wording correct.

Lines 259-260: The order in this sentence is nutrients and bioactives, but the following section is in reverse order. Please align.

Corrected (299-305). Thank you for identifying this gaffe.

Figure 3: The study represented by panel A used a combination of high-dosed nutrients and is not specific for lutein. Hence, it does not represent lutein intake and MPOD and should rather be removed.

The figure has been removed. In its place, we have a figure specific to lutein from Bone and Landrum, 2010 (citation 84) (331-338). Thank you for noting this error on my part.

Line 403-404: Note that soybeans contain only a modest amount of vitamin K as phylloquinone (K1). Natto contains menaquinone (K2) which is formed by bacteria during fermentation. It cannot be claimed that legumes in general are rich sources of MK-7.

Thank you for identifying this error. The section has been corrected to read “….and MK-7 is found in fermented soybeans (natto) where it is formed by bacteria during fermentation. Vitamin K is also produce by gut microbiota but their contribution to vitamin K status is unclear [69]”. (448-451).

Round 2

Reviewer 1 Report

The revised manuscript addressed most of my previous concerns.